# Essential Oils: A Natural Weapon against Antibiotic-Resistant Bacteria Responsible for Nosocomial Infections

**DOI:** 10.3390/antibiotics10040417

**Published:** 2021-04-10

**Authors:** Ramona Iseppi, Martina Mariani, Carla Condò, Carla Sabia, Patrizia Messi

**Affiliations:** Department of Life Sciences, University of Modena and Reggio Emilia, Via G. Campi 103/287, 41125 Modena, Italy; martina.mariani@yhaoo.com (M.M.); carla.condo@hotmail.it (C.C.); patrizia.messi@unimore.it (P.M.)

**Keywords:** antibiotic-resistant pathogens, essential oils, antibiotics, synergy association, anti-biofilm activity, MRSA, methicillin-resistant *Staphylococcus aureus*, VRE, vancomycin-resistant enterococci, ESBL, extended-spectrum β-lactamase *Escherichia coli*

## Abstract

The emergence of antibiotic-resistant bacteria has become a major concern worldwide. This trend indicates the need for alternative agents to antibiotics, such as natural compounds of plant origin. Using agar disc diffusion and minimum inhibitory concentration (MIC) assays, we investigated the antimicrobial activity of *Citrus aurantium* (AEO), *Citrus x limon* (LEO), *Eucalyptus globulus* (EEO), *Melaleuca alternifolia* (TTO), and *Cupressus sempervirens* (CEO) essential oils (EOs) against three representatives of antibiotic-resistant pathogens and respective biofilms: vancomycin-resistant enterococci (VRE), methicillin-resistant *Staphylococcus aureus* (MRSA), and extended-spectrum β-lactamase (ESBL)-producing *Escherichia coli*. Using the checkerboard method, the efficacy of the EOs alone, in an association with each other, or in combination with the reference antibiotics was quantified by calculating fractional inhibitory concentrations (FICs). All the EOs displayed antibacterial activity against all strains to different extents, and TTO was the most effective. The results of the EO–EO associations and EO–antibiotic combinations clearly showed a synergistic outcome in most tests. Lastly, the effectiveness of EOs both alone and in association or combination against biofilm formed by the antibiotic-resistant strains was comparable to, and sometimes better than, that of the reference antibiotics. In conclusion, the combination of EOs and antibiotics represents a promising therapeutic strategy against antibiotic-resistant bacteria, even protected inside biofilms, which can allow decreasing the concentrations of antibiotics used.

## 1. Introduction

In recent years, the frequency of bacteria resistant to multiple pharmacological agents has steadily increased [1]. The emergence of antibiotic-resistant bacteria has become a major concern worldwide, with considerable clinical and economic impact, as recently highlighted in the 2019 document by the WHO [2] that inserts the antibiotic resistance among the ten threats to global health. Antibiotic-resistant bacteria cause nosocomial infections [3] that are more difficult to treat and that can easily lead to death. In Europe, antibiotic-resistant pathogens cause more than 33,000 deaths every year [4], and about one-third of these occur in Italy. In fact, Italy is one of the European countries with the highest percentages of antimicrobial resistance in the main pathogens under surveillance (*Staphylococcus aureus*, *Enterococcus faecalis*, *Enterococcus faecium*, *Escherichia coli*, etc.), as well as the biggest consumer of antibiotics, both in human and veterinary fields. The increase in resistance to almost all classes of antibiotics and therefore the difficulty in treating an infection goes hand in hand with the need for new antibiotics and strategies to deal with this threat [5]. However, the discoveries and research in this field do not keep up in providing new and effective therapeutic agents. For this reason, the scientific community is currently showing a growing interest in the biodiversity of the world of plants and their derivatives, as they represent important sources of potential molecules with a broad spectrum of action. Many drugs, including antimicrobials, that have long been used in medicine contain one or more ingredients derived from plants or are synthetically developed from them. At the center of various investigations, essential oils (EOs), complex biochemical mixtures extracted from aromatic plants, have shown a wide spectrum of antimicrobial activity against human pathogens, both in planktonic and in sessile form [6,7,8,9,10,11]. The antimicrobial potential of EOs has long been known, and in-depth data on the components responsible for this activity and the mode of action are available. Notably, *Citrus* EOs have been recognized as antibacterial compounds, with limonene as the main component, followed by ß-myrcene, linalool, ß-pinene, and α-pinene [12], the last also being the most representative compound found in *Cupressus sempervirens* EO (CEO) [13]. The biological activity of *Eucalyptus globulus* EO (EEO), is due to the high content of 1,8-cineole, a monoterpene also present in *Melaleuca alternifolia* EO (TTO) together with terpinen-4-ol, and *a*-terpineol [14,15]. The main damage is the cell membrane disruption, also linked to the inhibition of efflux pumps responsible for antibiotic resistance in Gram-negative bacteria, but other modes of action have been widely reported, such as the inhibition of the peptidoglycan layer synthesis of bacterial cell walls by binding to PBPs for Gram-positive bacteria [16,17,18,19,20]. Their antibacterial activity has been recently studied against pathogenic bacteria frequently responsible for hospital-acquired infections (HAIs) [21,22,23,24,25]. Recent studies have also highlighted the synergistic action of EOs with other antimicrobial drugs and how these associations could be used to positively modulate the antimicrobial activity of some antibiotics against antibiotic-resistant bacteria, with a consequent decrease in the minimum effective dose of the drugs [26,27,28,29,30,31,32,33,34].

Therefore, these natural compounds could represent, alone or in combination, a valid weapon against diseases caused by antibiotic-resistant microorganisms, even in pathologies where the etiological agents are organized in biofilms, such as urinary tract infections (UTIs), catheter-associated infections, or cystic fibrosis exacerbated by an infection caused by *Pseudomonas aeruginosa*. In this structure, microorganisms find protection from different adverse conditions and show a 10 to 1000 times higher tolerance to antimicrobial agents than the same cells in planktonic form [35,36]. Moreover, the nearness in which the different microbial species are found favors the exchange through the conjugation mechanism of genes located at the plasmid level responsible for antibiotic resistance.

The aim of the present investigation was to evaluate the antimicrobial activity of five EOs (*Citrus aurantium*, *Citrus x limon*, *Eucalyptus globulus*, *Melaleuca alternifolia* (tea tree oil), and *Cupressus sempervirens*) in vitro against strains belonging to three important representatives of antibiotic-resistant bacteria: vancomycin-resistant enterococci (VRE), methicillin-resistant *Staphylococcus aureus* (MRSA), and extended-spectrum β-lactamase (ESBL) producing *Escherichia coli*. The activity of EOs was analyzed using them alone and in combination with each other. With the purpose to investigate if these plant natural products can modulate the drug resistance phenomenon in antibiotic-resistant strains, EOs were also studied in combination with traditional antibiotics to which the bacterial strains under examination proved to be resistant. Lastly, the activity of EOs and various mixtures was examined on the biofilm produced by these strains. 

## 2. Results

### 2.1. Antibacterial Susceptibility and Synergy Testing

For the eight VRE strains, the resistance to vancomycin was confirmed, and four had minimum inhibitory concentration (MIC) values of 512 µg/mL. All MRSA strains were oxacillin-resistant, in most cases with MIC value of 512 µg/mL, whereas more uniformly distributed MIC values, ranging from 8 to 64 µg/mL, emerged for all ESBL-producing *E. coli* (Table 1 and Table 2).

Figure 1a–d shows the antibacterial activity of EOs, verified using the agar well diffusion assay as a preliminary screening. TTO showed the widest inhibitory effect towards 42 out of 44 strains, with the greatest bacterial growth inhibition zones, ranging from 11 to 20 mm. CEO was inactive on most MRSA strains, while it gave better results toward ESBL *E. coli* and even better results against VRE, with growth inhibition zones ranging from 6 to 10 mm. On the other hand, EEO was more effective against MRSA and VRE strains, with inhibition zones ranging from 6 to 21 mm for 7 out of 9 strains and 5 out of 8 strains, respectively, but it was less effective against ESBL *E. coli*. *Citrus x limon* EO (LEO) was more active on ESBL *E. coli* strains than on the others, with inhibition zones ranging from 6 to 10 mm for 29.6% of ESBL strains, 33% of MRSA strains, and 8% of VRE stains and from 11 to 20 mm for the 26%, 11.1%, and 10%, respectively. Lastly, *Citrus aurantium* EO (AEO) proved to be the one with the least activity towards all the antibiotic-resistant strains (inhibition zones ranging from 6 to 21 mm for only one MRSA and one VRE, inhibition zones greater than 21 mm for only one ESBL).

The results of MIC determination (Table 3 and Table 4) confirmed TTO as the most effective EO, showing its activity against 25 out of 27 ESBL *E. coli*, most with low MIC ranging from 0.5 to 16 µg/mL. All VRE and MRSA were sensitive to TTO, even in this case with a low level of active substances (most with values equal to or less than 32 µg/mL). CEO was inactive towards the majority of MRSA strains, while it gave better results on ESBL-producing *E. coli* and even better results on VRE strains, with MIC values ranging from 32 to >512 µg/mL and from 32 to 256 µg/mL, respectively. EEO showed good activity against 7 out 9 MRSA strains (MIC from 8 to 128 µg/mL), but only 4 out of 27 ESBL-producing *E. coli* (MIC from 32 to 256 µg/mL) and 3 out of 10 VRE (MIC from 8 to 16 µg/mL) were sensitive. LEO was effective on 15 out of 27 ESBL-producing *E. coli* strains, but with high MIC values (MIC from 128 to >512 µg/mL), and 4 out of 9 MRSA and 2 out of 10 VRE were sensitive to this essential oil. Lastly, AEO showed the least activity, with only three out of all antibiotic-resistant strains being sensitive to this compound (MIC from 128 to 512 µg/mL for one VRE, one MRSA, and one ESBL strain).

Lastly, the results of the antibacterial activity of EO–EO associations and EO–antibiotic combinations clearly showed the synergistic effect in a large number of determinations of active compounds, while no antagonistic effects were found. Some of the most effective EO–EO associations and EO–antibiotic combinations are listed in Table 5. Notably, TTO and CEO proved to be the most potent EOs capable of having a synergistic effect against ESBL-producing *E. coli* strains, both when combined with the antibiotic and when associated with each other. With regard to VRE strains, the best synergy emerged using the TTO–VAN combination, followed by CEO–VAN, and this latter EO was already found to be the second most active compound after the TTO against VRE based on the MIC values. Even in this case, the TTO–CEO association proved to be the most frequent synergy. With respect to MRSA strains, the most evident synergies with the antibiotic were those relating to the TTO–OXA association (fractional inhibitory concentrations (FICs) 0.5, 0.04, and 0.03), followed by EEO–OXA (FIC 0.5). These two EOs, when in an association, also showed a synergistic effect against the MRSA strains.

Regarding EO–EO associations, it is interesting to note that when compounds that were less effective against most of the tested strains, such as AEO and LEO, were used together with the other compounds, they were shown to be effective at dilutions 3-fold less than the single used EO (Appendix A). The same advantage was observed in all the other synergies of EOs that were found to be effective when used alone (TTO–CEO, TTO–EEO) (Appendix A). These results are consistent with other investigations carried out combining different EOs in order to increase their antibacterial effect without increasing their concentration of use [37].

Table 6 shows some examples of combination or association compared to the MIC of the respective antibiotics (one for each genus). With rare exceptions, the antibiotic concentrations, when the compound was used together with the EO, were lower than the breakpoint of each species, showing a positive modulation in the reduction of the multidrug resistance among the pathogenic strains tested. The association of different EOs also led to an increase in their antibacterial effect, with a contextual decrease in their employed concentrations. 

### 2.2. Time–Kill Studies

The bacterial growth of three antibiotic-resistant strains (VRE B5, MRSA O, and ESBL *E. coli* 34), one chosen for each genus on the basis of the best synergies emerged in the previous determinations, was evaluated at 37 °C in optical density (OD) at 595 nm at selected time intervals (3, 6, and 24 h). The strains were exposed to antibiotic alone, EOs alone, and different synergistic mixtures of EO–antibiotic and EO–EO, as shown in Figure 2.

TTO showed the best inhibition of VRE B5 growth after 24 h of incubation, both alone and in the TTO–CEO association. As for the other synergies, the best results are mainly represented by the TTO–VAN and CEO–VAN combinations, with significantly lower usage concentrations than the antibiotic alone. 

For MRSA O, EO–EO associations and OE–antibiotic combinations led to the decrease in the microbial load. TTO–EEO association gave the best synergistic result, followed by TTO–CEO. Among the combinations with the antibiotic, TTO–OXA and EEO–OXA showed the greatest synergistic effect, with bacterial load reduction obtained at low concentrations of both synthetic and natural compounds. 

The activity of all essential oils, analyzed alone or in combination, towards ESBL *E. coli* 34 was confirmed, with few exceptions. TTO and CEO alone achieved the best results, inhibiting bacterial growth to very low OD values, followed by their combination with CTX. The synergistic effect highlighted above was also confirmed for the TTO–CEO association. 

### 2.3. EO Activity on Mature Biofilm 

To determine the anti-biofilm activity of EOs, the capability of the single strains to form biofilms was studied. The results showed that 13 strains of ESBL-producing *E. coli* (39, 22A, 26A, 28A, 38A, 36AT, 22BT, 23CT, 40CT, 24DT, 27G, 22F, and 31FT) were nonproducers at 24 h; therefore, they were excluded from the test. Among the producers, three antibiotic-resistant strains (ESBL *E. coli* 34, VRE B5 and MRSA O), one chosen for each genus on the basis of their sensitivity to the natural compounds used alone or in association, were grown alone (positive control), with the reference antibiotic, and with the EOs previously found to be the most active and to have best FIC index. Therefore, for ESBL *E. coli* 34, VRE B5, and MRSA O biofilms, TTO–CEO and TTO–EEO associations, together with their synergistic EO–antibiotic combinations, were employed.

As shown in Figure 3a–c, TTO tested alone and in combination with VAN caused the reduction of mature biofilm, as did the synergistic TTO–CEO association. The EOs tested alone and in association with each other and in combination with oxacillin resulted in the decrement of MRSA biofilm. The best results were obtained both with the synergistic combinations TTO–OXA and EEO–OXA and with the TTO–EEO association. The single EOs or the EOs in combination or association showed appreciable antibiofilm potential against the producer ESBL-producing *E. coli*. Notably, the presence of TTO led to a reduction of biofilm, as did the combinations of TTO–CTX and CEO–CTX. 

## 3. Discussion

One of the main causes of antibiotic resistance is the abuse of antibiotics for therapeutic purposes in both humans and the veterinary field. Furthermore, the bacteria have developed resistance to all classes of antibiotics, and there is a continuous need to produce new drugs. As reported in other investigations, essential oils represent an important source of compounds active against pathogens of public health interest, especially antibiotic-resistant strains. The EOs’ inhibition of bacterial growth could be due to the capacity to degrade membrane proteins and to increase cell permeability [38]. Moreover, EOs are active in the early stages of biofilm formation, inhibiting the adhesion of bacterial cells or interfering with quorum sensing [39], a system of bacterial communication that controls biofilm formation, antibiotic resistance, and expression of virulence factors.

In this study, the antibacterial activity of five different essential oils was evaluated against ESBL-producing *E. coli*, VRE, and MRSA clinical isolates, using these natural compounds alone, in association with each other, or in combination with the reference antibiotics. TTO showed the best bactericidal activity against all tested microorganisms, alone and in association with other EOs and antibiotics [40]; CEO was effective against ESBL *E. coli* and VRE, and EEO was effective against MRSA.

When EOs and antibiotics were tested in combination against antibiotic-resistant bacteria, a significant reduction in concentrations of the antibiotics was observed in many cases, with values much lower than the breakpoint set for that species, and these results were consistent with those of other authors [41,42,43,44]. The reason for the numerous synergistic effects that emerged both in the present study and in other investigations could be linked to the mechanism of action of EOs, in particular their ability to interact with the permeability of the bacterial cell in Gram-negative bacteria and their capability to alter the gene regulation involved in the cell wall metabolism in Gram-positive bacteria [45]. The interaction of EOs with the bacterial cell wall could be used to improve the activity of antibiotics, facilitating their penetration with a reduction of therapeutic doses [46,47,48].

An additive effect was found when EOs were combined with antibiotics and when EOs were combined with other EOs, especially when mixing TTO with EEO or CEO. For CEO, the synergistic effect was observed when it was combined with vancomycin and ceftazidime, reference antibiotics for VRE and ESBL-producing *E. coli*, respectively. EEO showed the best results on MRSA strains, mainly when used in combination with oxacillin and in association with TTO. LEO showed relevant activity alone and in combination with antibiotics and EOs on MRSA strains and showed even better results on ESBL *E. coli*, although with higher concentrations than TTO. On the contrary, AEO was found to be inactive on most of the bacteria tested and, like LEO, gave better results when mixed with other EOs, being effective at dilutions 3-fold less than when singularly used. 

Some EOs presented an anti-biofilm activity on both Gram-positive and Gram-negative pathogenic bacteria at the first stages of biofilm development and quorum sensing biosynthesis [49].

Lastly, these data could be of particular interest in contexts other than the clinical one, such as in food, where the typical aroma of essential oils could represent a problem from an organoleptic point of view. The association of one or more EOs that reduce MICs by 2–3 times compared to the single compound would therefore have a corrective effect on the smell problem, allowing the addition to food as preservatives without affecting its sensorial quality [50,51].

## 4. Materials and Methods 

### 4.1. Essential Oils

EO samples from *Citrus aurantium* subsp. *amara* Engler (AEO), *Citrus x limon* L. Osbeck (LEO), *Eucalyptus globulus* Labill. (EEO), *Melaleuca alternifolia* (Maiden and Betche) (tea tree) (TTO), and *Cupressus sempervirens* L. (Mill.) (CEO), all obtained by hydrodistillation, were purchased from a local herbalist’s shop in Modena, Italy.

### 4.2. Microbial Strains

All the strains, 27 ESBL-producing *E. coli*, 8 VRE, and 9 MRSA (Table 1 and Table 2), were isolated in the Provincial Laboratory of Clinical Microbiology “S. Agostino-Estense” Hospital (Modena, Italy), confirmed by matrix-assisted laser desorption ionization (MALDI) time-of-flight mass spectrometry (TOF-MS). The antimicrobial susceptibility testing was carried out using the Vitek 2 system (bioMerieux, Florence, Italy). All strains were maintained at −80 °C in media containing 20% (*w/v*) glycerol until use.

### 4.3. Antibacterial Susceptibility Testing

The preliminary determination of the activity of EOs against all the bacteria tested was carried out by using the agar disk diffusion assay, according to the standard procedure of the Clinical and Laboratory Standards Institute [52]. Tryptic soy agar (TSA, Oxoid) plates were seeded with 100 µL of 10^6^ CFU/mL of cell suspensions, and then sterile disks of 6 mm in diameter, containing 10 µL of each EO, were placed on these plates. After incubation at 37 °C for 24 h, the antibacterial activity of the EOs was quantified by a clear zone of inhibition in the indicator lawn around the disks, and the diameters in millimeters of these zones were measured [53].

According to the Clinical Laboratory Standards Institute (CLSI) guidelines (2019) [52], the MIC values of both antibiotics and EOs were determined against all microorganisms by the broth microdilution method in 96-well microplates, using oxacillin, vancomycin, and ceftazidime as antibiotics for MRSA, VRE, and ESBL *E. coli*, respectively. 

In each well of a sterile 96-well microplate, 95 µL of tryptic soy broth (Oxoid S.p.A, Milan, Italy) and 5 µL of bacterial suspensions were added, to a final inoculum concentration of 10^6^ CFU/mL. Then, 100 µL of EOs serial dilutions were added to obtain concentrations ranging from 512 to 0.125 μg/mL. Negative control wells consisted of bacteria in TSB without antibiotics and EOs. The plates were mixed on a plate shaker at 300 rpm for 20 s and incubated at 37 °C for 24 h. The MIC was defined as the lowest concentration of antibiotic or EO that inhibited visible growth of the tested microorganisms when the optical density (OD) was measured at 570 nm using a microtiter plate reader. All the experiments were conducted in triplicate, and results were expressed as the arithmetic mean of the three determinations.

### 4.4. Determination of the Fractional Inhibitory (FIC) Index—Synergistic Testing

Using the fractional inhibitory concentration (FIC) index, EO–EO associations and EO–antibiotic combinations were used to analyze their activity toward six antibiotic-resistant strains, chosen, two for each genus, on the basis of the results of the sensitivity studies. The antimicrobial assays were performed using the checkerboard method with a 96-well microplate [54], and the FIC index was calculated by comparing the value of the MIC of each agent alone with the combination-derived MIC. EOs were tested in the range of 1 MIC to 1/8 MIC in all possible variants. The results were classified as synergy (FIC ≤ 0.5), addition (0.5 ≤ FIC > 1), indifference (1 ≤ FIC > 4), and antagonism (FIC > 4). The experiments were conducted in the same manner as for the MIC determination in the susceptibility testing. 

### 4.5. Time–Kill Studies

Three antibiotic-resistant strains, one chosen for each genus on the basis of the best synergies that emerged in the previous determinations, were exposed to the corresponding antibiotics, EOs, and the different combinations of compounds (EO–EO and EO–antibiotic), and the bacterial load was determined by calculating the difference in the optical density. In a 96-well sterile microplate, 90 µL of sterile nutrient broth and 10 µL of the strains were placed in each well from a stock dilution to obtain a density of about 10^5^ CFU/mL. Antimicrobials and EOs were added at different concentrations, depending on the results obtained during MIC and the fractional inhibitory concentration index assays. The microplate was incubated at 37 °C with an oscillating speed of 150 rpm, and the optical density (OD) was determined at 595 nm at selected time intervals (0, 3, 6 and 24 h) of exposure, using an automatic microplate reader (Tecan Sunrise). The experiments were replicated three times.

### 4.6. EO Activity on Mature Biofilm

The capability of all antibiotic-resistant strains to form biofilm was studied using a modified 96-well microtiter plate method [55]. The effects of EOs, antibiotics, and the EO–EO and EO–antibiotic combinations on 24 h formed biofilm were evaluated according to an adapted Kwieciński et al. (2009) method [56]. For each microorganism, an overnight culture (18–24 h) was diluted in fresh sterile TSB to the final concentration of 10^6^ CFU/mL, and 150 μL was dispensed into each well of a 96-well plate. The biofilm production was evaluated at 37 °C after 24 h. A well containing only sterile TSB was used as negative control. Only the strains that were found to be the best biofilm producers after 24 h were selected to determine the activity of single compounds and their associations on biofilm.

After biofilm formation, to remove planktonic bacteria, the medium was gently aspirated, wells were washed three times with a sterile phosphate-buffered saline solution (PBS, pH 7.2), and the compounds were added at MIC and/or at the synergic concentration. Following an additional incubation at 37 °C for 24 h, according to the crystal violet staining method by Stepanovic et al. (2000) [55], the biofilm biomass was quantified.

After incubation, plates were washed three times with a sterile phosphate-buffered saline solution (PBS, pH 7.2) to remove planktonic bacteria and fixed with 150 μL of methanol for 15 min. Plates were then emptied, air-dried, and stained with 150 μL of crystal violet (0.2%, Hucker crystal violet) for 15 min at room temperature. After staining, wells were washed three times with sterile PBS, and the dye bound to the cells was dissolved in 33% glacial acetic acid (Sigma-Aldrich, Saint Louis, MO, USA). Results were expressed in terms of optical density (OD) values at 570 nm obtained using a microplate reader (Sunrise Tecan, Austria), as the arithmetic mean of the three determinations, and the standard deviation was reported as error bars.

### 4.7. Statistical Analysis

Each experiment was replicated three times. The statistical analysis was performed by *t*-test and ANOVA test. The *p*-values were considered statistically significant at ≤0.05.

## 5. Conclusions

The emergence of antibiotic-resistant bacteria has become a major problem around the world and has had a significant clinical and economic impact. Within this scenario arose the idea to use natural substances capable of positively modulating the sensitivity of antibiotic-resistant pathogens even when they are organized in biofilms. Our data highlight the ability of EOs to represent a valid alternative as a means of dealing with the problem of antibiotic-resistant pathogens. Furthermore, the use of a combination of EOs with antibiotics could represent a promising strategy to counter antibiotic-resistant bacterial infections, even when protected by biofilm structures. This strategy would also allow the concentrations of antibiotics used to be decreased, with a consequent reduction in the spread of drug resistance. Moreover, our results underline that EOs could have important applications in the discovery and implementation of new antimicrobial strategies in other fields besides therapeutic use, such as cosmetics, food, and environmental fields. 

## Figures and Tables

**Figure 1 antibiotics-10-00417-f001:**
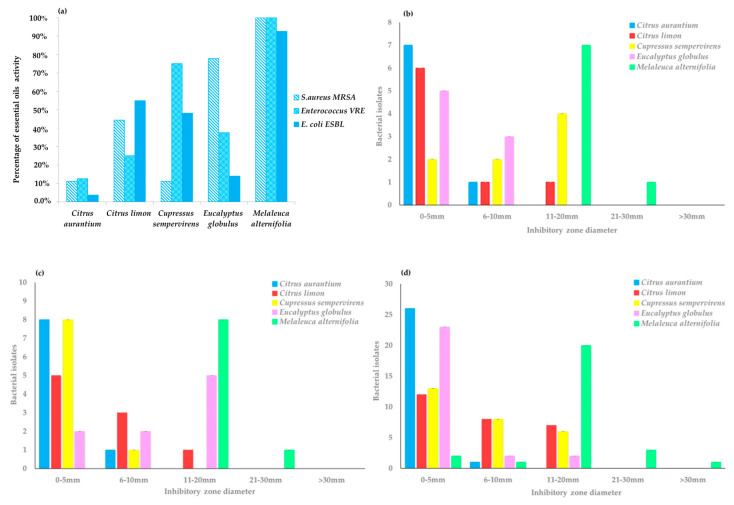
Antibacterial activity of essential oils (EOs) verified using the agar well diffusion method. Percentage of activity against tested antibiotic-resistant strains (**a**). Ranges of inhibitory zone diameter for VRE (**b**), MRSA (**c**), and ESBL-producing *E. coli* (**d**) strains.

**Figure 2 antibiotics-10-00417-f002:**
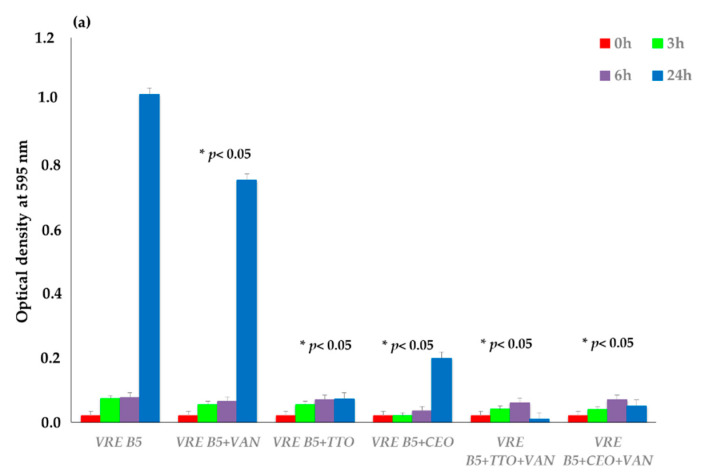
Time–kill studies of EOs, antimicrobials (VAN, OXA, and CTX), and the different combinations (EO–EO, EO–antimicrobial) against VRE B5 (**a**), MRSA O (**b**), and ESBL *E. coli* 34 (**c**) strains. *p*-values of <0.05 (*), <0.01 (**) were considered significant by *t*-test and ANOVA.

**Figure 3 antibiotics-10-00417-f003:**
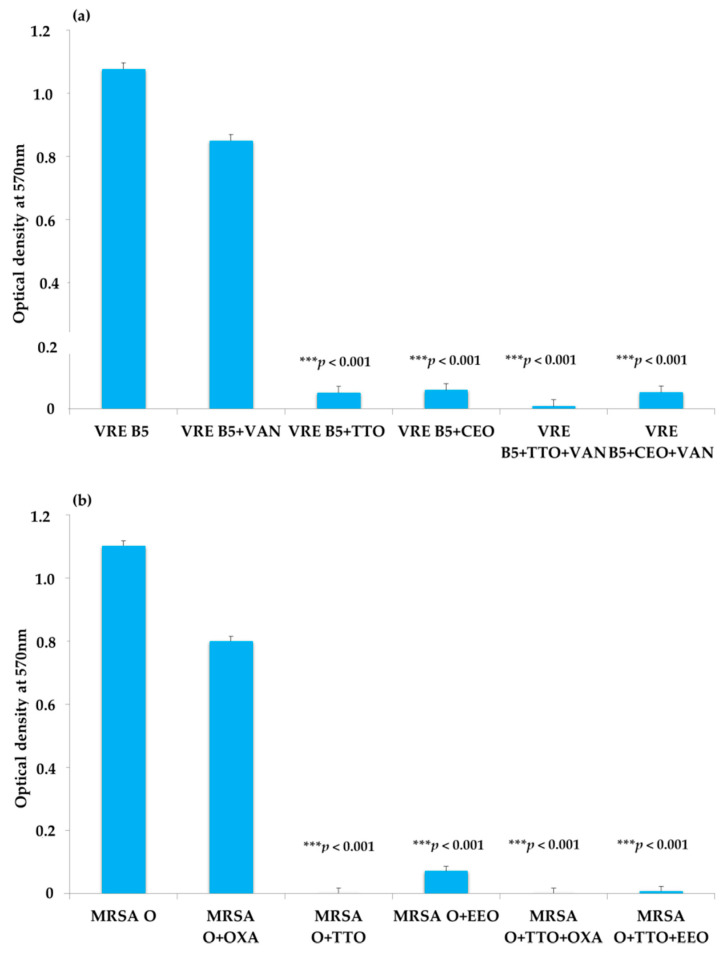
Effect of selected EOs, alone and in combination with reference antibiotic, on mature biofilm formed by VRE B5 (**a**), MRSA O (**b**), and ESBL *E. coli* 34 (**c**) strains. *p*-values of <0.05 (*), <0.01 (**), <0.001 (***) were considered significant by *t*-test and ANOVA.

**Table 1 antibiotics-10-00417-t001:** Minimum inhibitory concentration (MIC) of vancomycin and oxacillin (µg/mL) against vancomycin-resistant enterococci (VRE) and methicillin-resistant *Staphylococcus aureus* (MRSA) strains, respectively. Abbreviations for antibiotics: VAN, vancomycin; OXA, oxacillin. Abbreviations for interpretations: S = sensitive; I = intermediate; R = Resistant.

Strains	VAN		Strains	OXA	
*E. faecium* A29	512	R	*S. aureus* 6	512	R
*E. faecium* A30	512	R	*S. aureus* 12A	512	R
*E. faecium* B5	512	R	*S. aureus* 12B	512	R
*E. faecium* VAN 2	16	R	*S. aureus* C1	512	R
*E. faecalis* VAN 3	128	R	*S. aureus* C3	512	R
*E. faecalis* VAN 4	512	R	*S. aureus* MRSA	512	R
*E. faecium* VAN 5	8	R	*S. aureus* MRSA1	8	R
*E. faecalis* VAN 19	8	R	*S. aureus* MRSA2	8	R
			*S. aureus* O	512	R

**Table 2 antibiotics-10-00417-t002:** Minimum inhibitory concentration (MIC) of cefotaxime (mcg mL^−1^) against extended-spectrum β-lactamase (ESBL)-producing *E. coli* strains. Abbreviations for antibiotic: CTX, cefotaxime. Abbreviations for interpretations: S = sensitive; I = intermediate; R = Resistant.

Strains	CTX		Strains	CTX	
*E. coli* 34	32	R	*E. coli* 23CT	64	R
*E. coli* 39	64	R	*E. coli* 40CT	32	R
*E. coli* 22A	64	R	*E. coli* 23DT	64	R
*E. coli* 26A	32	R	*E. coli* 24DT	64	R
*E. coli* 28A	32	R	*E. coli* 31DT	32	R
*E. coli* 38A	32	R	*E. coli* 45DT	32	R
*E. coli* 41A	32	R	*E. coli* 22F	64	R
*E. coli* 24AT	64	R	*E. coli* 23F	64	R
*E. coli* 36AT	32	R	*E. coli* 31FT	64	R
*E. coli* 24B	64	R	*E. coli* 27G	64	R
*E. coli* 22BT	64	R	*E. coli* 4 CL	64	R
*E. coli* 36BT	64	R	*E. coli* 9 CL	16	R
*E. coli* 24C	8	R	*E. coli* 11 CL	64	R
*E. coli* 22CT	64	R			

**Table 3 antibiotics-10-00417-t003:** Minimum inhibitory concentration (MIC) of EOs (µg/mL) against VRE and MRSA strains.

Strains	AEO	LEO	EEO	TTO	CEO	Strains	AEO	LEO	EEO	TTO	CEO
*E. faecium* A29	256	64	8	1	32	*S. aureus* 6	>512	>512	128	32	>512
*E. faecium* A30	>512	>512	8	16	128	*S. aureus* 12A	>512	32	64	32	>512
*E. faecium* B5	>512	>512	>512	8	64	*S. aureus* 12B	>512	256	>512	64	>512
*E. faecium* VAN2	>512	>512	>512	128	>512	*S. aureus* C1	>512	>512	8	4	>512
*E. faecalis* VAN3	>512	256	16	64	64	*S. aureus* C3	>512	>512	32	8	>512
*E. faecalis* VAN4	>512	>512	512	8	128	*S. aureus* MRSA	128	32	32	8	>512
*E. faecium* VAN5	>512	>512	>512	32	>512	*S. aureus* MRSA1	>512	>512	>512	32	512
*E. faecalis* VAN 19	>512	>512	>512	64	256	*S. aureus* MRSA2	>512	32	8	16	>512
						*S. aureus* O	>512	>512	32	8	>512

**Table 4 antibiotics-10-00417-t004:** Minimum inhibitory concentration (MIC) of EOs (µg/mL) against ESBL-producing *E. coli* strains.

Strains	AEO	LEO	EEO	TTO	CEO	Strains	AEO	LEO	EEO	TTO	CEO
*E. coli* 34	>512	256	>512	8	64	*E. coli* 23CT	>512	>512	>512	2	>512
*E. coli* 39	>512	>512	>512	>512	32	*E. coli* 40CT	>512	256	>512	4	>512
*E. coli* 22A	>512	512	>512	8	32	*E. coli* 23DT	>512	>512	>512	2	>512
*E. coli* 26A	>512	>512.	>512	1	>512	*E. coli* 24DT	>512	>512	>512	0.5	>512
*E. coli* 28A	>512	256	64	4	64	*E. coli* 31DT	>512	512	>512	1	64
*E. coli* 38A	>512	256	>512	4	128	*E. coli* 45DT	>512	>512	128	1	128
*E. coli* 41A	>512	256	>512	4	>512	*E. coli* 22F	>512	512	>512	2	64
*E. coli* 24AT	>512	>512	>512	4	>512	*E. coli* 23F	>512	128	>512	1	>512
*E. coli* 36AT	>512	>512	>512	2	>512	*E. coli* 31FT	>512	512	256	16	>512
*E. coli* 24B	>512	>512	>512	0.5	>512	*E. coli* 27G	>512	>512	>512	2	64
*E. coli* 22BT	>512	256	32	128	128	*E. coli* 4 CL	>512	>512	>512	2	>512
*E. coli* 36BT	>512	>512	>512	1	>512	*E. coli* 9 CL	>512	>512	>512	4	>512
*E. coli* 24C	>512	>512	>512	256	>512	*E. coli* 11 CL	512	>512	>512	>512	>512
*E. coli* 22CT	>512	256	>512	4	64						

**Table 5 antibiotics-10-00417-t005:** Examples of synergistic activity of EO–EO associations and EO–antibiotic combinations against ESBL-producing *E. coli*, VRE, and MRSA strains, by fractional inhibitory concentration (FIC) index calculation.

Strains	TTO/CTX	CEO/CTX	TTO/CEO	Strains	TTO/VAN	CEO/VAN	TTO/CEO
*E. coli* 34	0.04	0.5	0.5	*E. faecium* A29	0.25	0.5	0.5
*E. coli* 22A	0.27	0.5	0.5	*E. faecium* A30	0.02	0.5	1.5
*E. coli* 28A	0.07	0.5	0.5	*E. faecium* B5	0.03	0.5	1.5
*E. coli* 38A	0.07	0.5	0.5	*E. faecalis* VAN4	0.5	0.5	1.5
*E. coli* 22BT	0.5	0.5	0.5				
*E. coli* 23DT	0.27	0.5	0.5	**Strains**	**TTO/OXA**	**EEO/OXA**	**TTO/EEO**
*E. coli* 31DT	0.26	0.5	1.5	*S. aureus* 12A	0.5	0.5	0.5
*E. coli* 22F	0.13	0.5	0.5	*S. aureus* C3	0.03	0.5	0.5
*E. coli* 27G	0.14	0.5	2.0	*S. aureus* MRSA	0.04	2.0	0.5
*E. coli* 9 CL	0.27	0.5	0.5	*S. aureus* O	0.03	1.5	0.375

**Table 6 antibiotics-10-00417-t006:** Examples of synergy between MIC (µg/mL) of antimicrobials alone and of EO–EO association and EO–antibiotic combination against antibiotic-resistant pathogens (one for each genus).

Strains	MIC Alone	MIC Combined	FIC (A)	MIC Alone	MIC Combined	FIC (B)	FICI
VRE B5	TTO	TTO	TTO	VAN	VAN	VAN	0.032
	8	0.25	0.03	256	0.5	0.002	
	CEO	CEO	CEO	VAN	VAN	VAN	0.5
	64	16	0.25	256	64	0.25	
MRSA O	TTO	TTO	TTO	OXA	OXA	OXA	0.032
	8	0.25	0.03	256	0.5	0.002	
	EEO	EEO	EEO	TTO	TTO	TTO	0.375
	32	8	0.25	8	1	0.125	
ESBL *E. coli* 34	TTO	TTO	TTO	CTX	CTX	CTX	0.038
	8	0.25	0.03	32	0.25	0.008	
	CEO	CEO	CEO	CTX	CTX	CTX	0.5
	64	16	0.25	32	8	0.25	
	TTO	TTO	TTO	CEO	CEO	CEO	0.5
	8	2	0.25	64	16	0.25	

## Data Availability

Not applicable.

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
