# Peer review of "Essential Oils: A Natural Weapon against Antibiotic-Resistant Bacteria Responsible for Nosocomial Infections"

_antibiotics, 2021, doi:10.3390/antibiotics10040417_

Round 1
Reviewer 1 Report
Presented manuscript – “ Essential oils: a weapon against MDR bacteria responsible for nosocomial infections” responds to the pressing problem of global spread of antibiotic resistance among bacteria, and is an important contribution to the search for new agents that could be effectively used in the fight against infections with multi-drug-resistant strains.
Extensive and detailed multi-strain research has been carried out and appropriate research methodology has been applied. Interesting and promissing results were obtained and presented clearly in the form of graphs and tables
I have two comments about the presented work:
a) there is no results of resistance of the tested bacterial strains to antibiotics other than those used in combination with EO . Perhaps a summary of the “full” drug resistance according to the division into classes of antibiotics in terms of their chemical structure and mechanisms of action would shed new light on the results obtained, for example, the lack of resistance of some strains to essential oils tested and the synergy effect.
b) In the Introduction section, there is no information on the mechanisms of action of essential oils on bacterial cells, taking into account the division into Gram positive and Gram negative cells , and a discussion of the obtained results in this context.
Reviewer 2 Report
The main aim of the original article entitled „Essential Oils: A Natural Weapon against MDR Bacteria Responsible for Nosocomial Infections” was to show the usefulness of selected essential oils against antibiotic-resistant strains of E. coli, Enterococcus and S. aureus.
In general, I think that the article is interesting and the results presented have applicative potential. Nevertheless, I have noticed a lot of mistakes in the text that reduce the quality of the manuscript and therefore, in my opinion, need to be corrected. In the future, I suggest to pay more attention to details, as they often ultimately determine the receipt of the manuscript.
The list of suggested corrections:
- at the very beginning, I believe that the text requires more thorough correction by an English native speaker -> hence, I am asking you to improve the linguistic quality of the text
- "MDR" should be removed from the manuscript title and ALL other places in the text and replaced by “antibiotic-resistant” -> the bacterial strains used in these studies had only a single resistance phenotype (resistance to beta-lactams or glycopeptides), while the MDR phenotype is defined as resistance to three or more groups of antibiotics [alternatively, if such resistance exists, it must be introduced in details in the methodology and results as it provides such important information to be even in the manuscript title]
- in the abstract, after the first sentence, please add a sentence that will bring the text together (currently, the mental jump can be observed), e.g., hence the high need for alternative methods ...
- “against 3 group of MDR pathogen” -> against three representatives of antibiotic-resistant pathogens [line 12]
- “methicillin-resistant (MRSA) Staphylococcus aureus” -> methicillin-resistant Staphylococcus aureus (MRSA) [line 13]
- “the efficacy of the essential oils (EOs)” -> the efficacy of the Eos [line 15]
- “and fractional inhibitory concentrations (FIC) calculated” -> calculating fractional inhibitory concentrations (FICs) [line 16]
- “In conclusion, the combination of EOs and antibiotics represents a promising therapeutic strategy against MDR bacteria, even protected inside biofilms, which can allow to decrease the concentrations of antibiotic used, thus reducing both the drug resistance spread and the costs of treatment.” -> this is not featured in the article, please change it [lines 21-24]
- “In recent years, bacteria resistant to …” -> In recent years, the frequency of bacteria resistant to … [line 30]
- “with additive and synergistic effects with antibiotics in use.” -> please delete this part, not useful here [line 52]
- “the synergistic role of EOs” -> the synergistic action of Eos [line 54]
- “(UTIs), in catheterized patients or cystic fibrosis by Pseudomonas aeruginosa.” -> (UTIs), catheter-associated infections or cystic fibrosis exacerbated by an infection caused by Pseudomonas aeruginosa. [line 61]
- “show a resistance to antimicrobial agents” -> show tolerance to antimicrobial agents [line 62]
- “The aim of the present …” -> please add a paragraph here
- “three main categories of” -> three important representatives of [line 70]
- “The activity of essential oils was analyzed …” -> The activity of EOs was analyzed … [line 71]
- “… the biofilm produced by the same strains” -> … the biofilm produced by these strains [line 76]
- “Eight out of ten VRE strains confirmed the resistance to vancomycin …” -> I don't understand this sentence, if VRE was detected in 8/10 of enterococci then the other two strains are just sensitive (the “vancomycin-intermediate enterococci” phenotype is not existing) -> hence, my question is: why are they included in this article? I believe they should be removed from the manuscript [line 79]
- “ … with a MIC values of 512 µg/mL” -> had MIC values of 512 µg/mL [line 80]
- “oxacillin resistant” -> oxacillin-resistant [line 80]
- “with MIC value of 512 µg/mL” -> with MIC values of 512 µg/mL [line 81]
- “growth inhibition zone” -> growth inhibition zones [line 93, 95, 97 and 99]
- “7 out 9 indicators and 7 out 10 indicators, respectively” -> 7 out 9 and 7 out 10 strains, respectively [lines 97-98]
- Figures -> I believe that all charts should be made in one style (Figure 1 and 2 differs from Figure 3), in addition, I ask you to change the color intensity of Figure 1 and 2, because they are very bright and after printing you can hardly see anything
- “Lastly, AEO proved to be the one with the least activity towards all the MDR strains.” AND “Lastly, AEO showed the least activity, with only 3 out all MDR strains sensitive to this compound.” -> please provide the value in brackets as previously described in the text [lines 100-101 and 116-117]
- “Percentage of activity against target …” -> Percentage of activity against tested [line 103]
- “All VRE and MRSA resulted sensitive …” -> All VRE and MRSA were sensitive … [line 107]
- “n.a. = not active.” under Table 3 and 4 -> "n.a." must have a value, e.g., > XXX µg/ml (being the peak value that has not been determined in testing anymore). Currently, in combination with > 512 µg/ml present in Tables (also an undefined value) it cannot be understood
- “n.a. = not active.” under the Table 5 -> the same as previously; interactions always have values, e.g. 2 being neutral or > 4 being antagonistic
- “TTO and CEO proved to be the main EOs capable of determining a synergistic effect” -> TTO and CEO proved to be the most potent EOs capable of having a synergistic effect [line 129]
- “With respect to MRSA strains, the most evident synergies with the antibiotic were those relating to the TTO / OXA association, followed by EEO / OXA.” -> it is worth providing FICI values in parentheses [lines 134-136]
- “… effective against quite all strains” -> effective against most of the tested strains [line 144]
- “was evaluated at 37 °C in optical density (OD) at 595 nm at defined time intervals” -> was evaluated at selected time intervals [line 164]
- “The strains were put in contact with the corresponding antibiotic, with the EOs and with different synergistic mixtures of antibiotic-EO and EO-EO, as shown in figure 2.” -> The strains were exposed to antibiotic alone, EOs alone, and with different synergistic mixtures of antibiotic-EO and EO-EO, as shown in the Figure 2. [line 165-167]
- “in the association TTO/CEO” -> in the TTO/CEO association [line 173]
- “The results showed that 13 strains of ESBL-producing coli (39, 22A, 26A, 28A, 38A, 36AT, 22BT, 23CT, 40CT, 24DT, 27G, 22F, 31FT) were non-producers” -> very unusual that so many strains of E. coli are not biofilm producers - this feature is not found frequently, especially in such vital bacteria as E. coli -> please explain [line 188]
- “with the reference antibiotic (negative control)” -> it is not a negative sample but rather an independent test sample; a negative sample would be a well containing only the culture medium [line 192]
- “As shown in figure 4 (a,b,c,) …” -> As shown in the Figure 4 (a,b,c,) … [line 196]
- Figure 3 -> Have the OD of the negative control been subtracted? There is no such information anywhere
- The discussion is far too short. The obtained results were not related to the literature data. Please, deepen the argument on the potential mechanism of the EOs activity and consider why there positively interact with antibiotics. In the context of an anti-biofilm activity, it is worth paying attention to the anti-QS effect of EOs
- “in the same media containing” -> in media containing [line 252]
- “the antagonistic activity of the EOs” -> the antibacterial activity of the EOs [line 259]
- “lawn around the disks” -> sometimes authors use "discs" in the text and sometimes "disks", please standardize [line 260]
- “using as antibiotics Oxacillin, Vancomycin and Ceftazidime” -> it is a pity that only cell wall-targeting antibiotics were used in the research; interesting results could be obtained using a different type of antimicrobials (worth considering in the future) [line 264-265]
- “EOs were combined at MIC + MIC, MIC + 1/2 MIC, MIC + 1/4 MIC, MIC + 1/8 MIC, 1/2 MIC + 1/2 MIC, 1/2 MIC + 1/4 MIC, 1/2 MIC + 1/8 MIC, 1/4 MIC + ¼ MIC, 1/4 MIC + 1/8 MIC, and 1/8 MIC + 1/8 MIC.” -> this is difficult to read; maybe: EOs were tested in the range of MIC-1/8 MIC in all possible variants. [lines 283-285]
- “were put in contact with the corresponding antibiotics” -> were exposed to the corresponding antibiotics [line 291]
- “the optical density (OD) was determined at 595 nm” -> Why 595 nm was used here when in other assays 570 nm was chosen? [line 299]
- “Sterile TSB was used as blank control.” -> It's an error. A negative control should be a well containing only culture medium, not an empty well as presented. This difference in methodology most likely influences the obtained results. [line 309]
- Conclusions are too long - these should be 2-4 sentences summarizing the obtained results (some part of the text can be used in the discussion)
- “data point in this directio” -> data point in this direction [line 338]
Reviewer 3 Report
The MS 1161365 entitled “Essential Oils: A Natural Weapon Against MDR Bacteria Responsible for Nosocomial Infections ”, by Iseppi et al., deals with the antimicrobial activity of a selection of essential oils (used as such, in association with each other and in combination with antibiotics) against MDR pathogens (i.e. VRE, MRSA, ESBL).
In my opinion, the work is well designed, well-structured in all its parts, the conclusions are opportunely argued and supported by the data; finally, it introduces novelties in the antimicrobial activities of essential oils against sessile and planktonic form of MDR bacteria. The work can therefore be published after few minor revision on Antibiotics.
Minor revision:
Please substitute “Citrus limon” (Line 11 and 243) with “Citrus x limon”
Please report the botanical authors after the Latin names of the plants in the 4.1 paragraph.
Round 2
Reviewer 2 Report
I would like to thank the Authors of the article for adapting to the suggestions of the Reviewers. I believe that the quality of the article has improved significantly and therefore I conclude in favor of publishing the manuscript.